# Using Membership Inference Attacks to Evaluate Privacy-Preserving Language Modeling Fails for Pseudonymizing Data

**Thomas Vakili and Hercules Dalianis**
Department of Computer and Systems Sciences (DSV)
Stockholm University, Kista, Sweden
{thomas.vakili,hercules}@dsv.su.se

## Abstract

Large pre-trained language models dominate the current state-of-the-art for many natural language processing applications, including the field of clinical NLP. Several studies have found that these can be susceptible to privacy attacks that are unacceptable in the clinical domain, where personally identifiable information (PII) must not be exposed.

However, there is no consensus regarding how to quantify the privacy risks of different models. One prominent suggestion is to quantify these risks using membership inference attacks. In this study, we show that a state-of-the-art membership inference attack on a clinical BERT model fails to detect the privacy benefits of pseudonymizing data. This suggests that such attacks may be inadequate for evaluating token-level privacy preservation of PIIs.

## 1 Introduction

State-of-the-art results in natural language processing typically rely on large pre-trained language models (PLMs) such as BERT (Devlin et al., 2019) or models in the GPT family (Radford et al., 2019). Multiple studies have found that their large number of parameters can cause PLMs to unintentionally memorize information in their training data, making them vulnerable to privacy attacks (Carlini et al., 2019, 2021). At the same time, other studies have shown that training PLMs using domain-specific data yields better results on domain-specific tasks (Lee et al., 2020; Lamproudis et al., 2021). In the clinical domain, these combined findings pose a significant challenge: training PLMs with clinical data is necessary to achieve state-of-the-art results. However,

PLMs can be vulnerable to privacy attacks that are especially dangerous when training with clinical data. Broadly speaking, these attacks can be divided into two classes: training data extraction attacks and membership inference attacks.

### 1.1 Privacy Attacks

Training data extraction attacks are the more severe class of attacks. An adversary who successfully mounts such an attack can extract details about training data that were used to train a PLM. Carlini et al. (2021) show that GPT-2 is vulnerable to such attacks. Several studies (Nakamura et al., 2020; Lehman et al., 2021; Vakili and Dalianis, 2021) have tried to mount similar attacks on BERT models. To this date, there are no examples of successful training data extraction attacks targeting BERT models.

Membership inference attacks (MIAs) do not aim to *extract* training data from models. Instead, these attacks try to discern whether or not a datapoint was present in a model's training data. Inferring that a datapoint has been present in the training data is less severe than extracting it but could, for example, reveal if a patient has visited a set of clinics.

MIAs have been proposed as a proxy for measuring the degree of memorization in machine learning models (Shokri et al., 2017; Murakonda and Shokri, 2020; Mireshghallah et al., 2022). Both training data extraction attacks and MIAs rely on some degree of memorization in the model. However, MIAs do not require any algorithms that generate the memorized data. By focusing solely on detecting memorization, MIAs are used to estimate a worst-case degree of privacy leakage. Indeed, MIAs are the basis for the ML Privacy Meter developed by Murakonda and Shokri (2020).

## 1.2 Protecting Datapoints or Tokens?

One special property of natural language data is that many words in a sentence can be replaced with synonyms without changing the overall semantics of the sentence. This feature is interesting from a privacy perspective and is the basis for *pseudonymization*.

Pseudonymization is the process of replacing sensitive information with realistic surrogate values. For example, names are replaced with other names or with placeholders. These kinds of sensitive words or phrases are rarely important for the utility of the data, neither for fine-tuning models (Berg et al., 2021; Vakili and Dalianis, 2022), pre-training models (Verkijk and Vossen, 2022; Vakili et al., 2022), nor for general research purposes (Meystre et al., 2014a,b). One important example of this is MIMIC-III (Johnson et al., 2016), which contains a large number of electronic health records in which sensitive words or phrases have been manually replaced with placeholders. This dataset is widely employed in clinical machine learning and is considered to be relatively safe.

One fundamental assumption of pseudonymization is that the higher-level semantics of a text are not important from a privacy perspective. For example, an electronic health record describing a patient visiting a hospital is not sensitive if we cannot infer *who* the patient is, *when* the visit took place, and so on. One way of viewing this is that the data are not primarily sensitive on the datapoint level, but on the token level.

## 1.3 Membership Inference Attacks and Pseudonymization

Manual pseudonymization is a time-consuming process. Many institutions lack the resources to manually pseudonymize data on the scale required for modern machine learning models or even for less data-intensive qualitative clinical research. An alternative is to use *automatic pseudonymization*. Automatic pseudonymizers typically rely on named entity recognition (NER) to detect sensitive information. The detected entities are then either replaced with realistic surrogates or with placeholders. However, NER systems are rarely perfectly accurate. Imperfect recall leads to some sensitive entities remaining after processing the data, which is undesirable from a privacy perspective.

Because systems performing automatic pseudonymization fail to detect some sensitive entities, it is important to measure the privacy implications of this. A straightforward approach is to consider the recall of the NER model that powers the system. This metric can be used to estimate the number of sensitive entities that remain in the data. Such estimates are useful for determining the sensitivity of an automatically pseudonymized dataset. However, they are less ideal for judging the privacy risks of a machine learning model trained using the dataset. Assuming that the trained model has memorized every single sensitive entity is overly pessimistic.

Estimating the privacy risks of models using MIA, as suggested by Mireshghallah et al. (2022), is an attractive alternative that would allow pseudonymization to be compared to other privacy-preserving techniques. However, MIAs are designed to measure the memorization of entire datapoints rather than the memorization of sensitive tokens. This poses a challenge to the paradigm of using MIAs to estimate the privacy risks of machine learning models trained using pseudonymized data.

In this study, we show that the state-of-the-art MIA described by Mireshghallah et al. (2022) cannot distinguish between a model trained using real or pseudonymized data. These results suggest that using this attack to quantify privacy risks fails to capture privacy gains from pseudonymizing training data.

## 2 Methods and Data

This study closely mirrors the experimental setup used by Mireshghallah et al. (2022) in order to minimize discrepancies stemming from differences in implementation details. The datasets and models are based on resources introduced by Lehman et al. (2021). The experiments aim to examine whether or not membership inference attacks can distinguish between a model trained using real or pseudonymized data.

### 2.1 Data

This study uses the ClinicalBERT-1a model trained by Lehman et al. (2021). They train a model using pseudonymized clinical notes from a subset of MIMIC-III. This specific model is of the same size as BERT-base (Devlin et al., 2019) and uses this model's parameters as a starting point for continued pre-training to adapt the model to the clinical domain. The corpus used to train

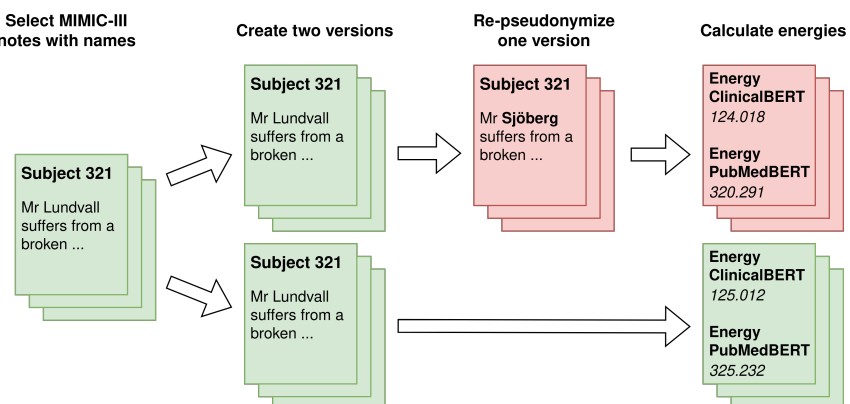

Figure 1: Our experiments use a filtered subset of MIMIC-III that only contains records with named (but pseudonymized) patients. One subset, the *Pseudo* subset, has been used to create the Clinical-BERT model used as the target for the attack. Another version, referred to as the *Real* dataset, is re-pseudonymized and acts as a stand-in for the original sensitive raw data.

the model is also available. Mireshghallah et al. (2022) perform their membership inference experiments using the training data for the BERT model and MIMIC-III data that was not used for training the model. The method also needs a reference model, and this study follows their example by also using PubMed-BERT (Gu et al., 2021) for this purpose.

This study focuses specifically on MIAs' ability to discern whether or not a model has been trained using pseudonymized data. A filtered version of MIMIC-III containing only sentences with names is created to ensure that the results reflect this distinction. This dataset contains a total of 236,114 datapoints. A pseudonymized version of the dataset is created in which all names have been replaced with other names.

After replacing all the names, we have two datasets where each sentence differs solely in what names are used. The dataset used to train the model will be referred to as the *Pseudo* dataset, and the re-pseudonymized dataset will be referred to as the *Real* dataset. This mimics the situation where we have a model trained on *perfectly pseudonymized* training data. Figure 2 illustrates the scenario that is simulated. Ideally, the membership inference attack should indicate that replacing all names with pseudonyms has made the model much safer.

## 2.2 Predicting Membership

This study uses the same procedures as Mireshghallah et al. (2022) since their method is the current state-of-the-art membership inference

attack targeting masked language models like BERT. The method works by analyzing how the target model reacts to a datapoint as compared to a reference model. The target and reference models, in our case ClinicalBERT and PubMed-BERT, differ in that the target model has been trained using sensitive data that the reference model has not been exposed to.

A variety of different measurements can represent the reaction of the model. Following the example of Mireshghallah et al. (2022), we use the *normalized energy values* calculated for every datapoint. These values $E_\theta(S)$ are calculated by estimating the probability of a sequence of tokens $S$ given a set of masking patterns $M$ for a model with the parameters $\theta$:

$$E_\theta(S) = \frac{1}{|M|} \sum_{m \in M} e_\theta(S, m)$$

$$e_\theta(S, m) = \sum_{i \in m} \log\left[p_\theta(S_i \mid S_m)\right]$$

$S_i$ is the token at index $i$ and $S_m$ is the altered sequence $S$ to which the masking pattern $m$ has been applied. These normalized energy values are calculated for three datasets, for both the target model and the reference model:

**In-data** Parts of the dataset used to train the target model. In this study, the two datasets described in Section 2.1 fill this function, as illustrated in Figure 1.

**Out-data** A second dataset known *not* to belong to the target models training data. This subset

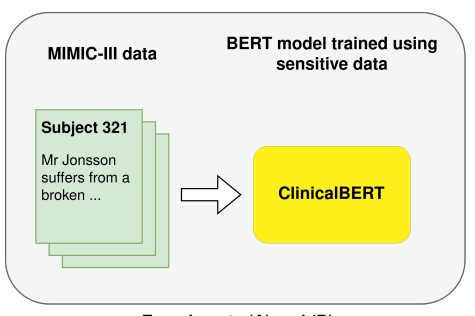 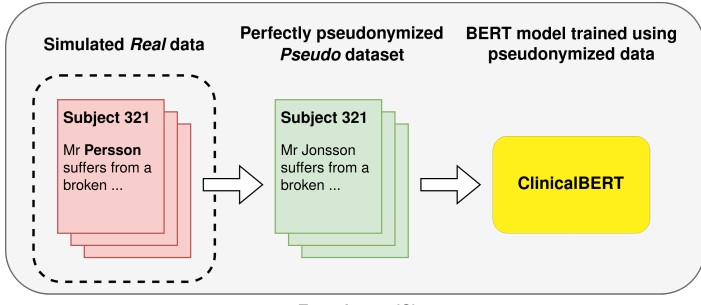

Experiments (A) and (B)        Experiment (C)

Figure 2: This study simulates the scenario in which a perfectly pseudonymized dataset has been used for continued pre-training of a BERT model. The version of MIMIC-III used to create the ClinicalBERT model from Lehman et al. (2021) is re-identified with pseudonyms and is used in experiments (A) and (B). We refer to this dataset as the *Pseudo* dataset. In experiment (C), we simulate the original, pre-pseudonymized MIMIC-III by populating the data with other names and call this version the *Real* dataset.

of MIMIC-III is also used in Mireshghallah et al. (2022).

**Threshold data** A third dataset disjoint from the *Out-data* and known not to belong to the target models training data. A subset of i2b2 (Stubbs and Uzuner, 2015) is used, as in Mireshghallah et al. (2022).

The normalized energy values of the target and reference models are compared for the threshold data, resulting in a threshold. This threshold is used to classify if a datapoints belongs to the *In-data* or the *Out-data* based on the difference between the energy values of the datapoint obtained from the models. The intuition behind this method is that if the target model has memorized a datapoint, then its energy value will be noticeably higher relative to the reference model's energy value. The threshold is set so that 90% of the datapoints in the *threshold dataset* are correctly classified as non-members (Mireshghallah et al., 2022). We also calculate the AUC to provide a threshold-independent assessment of the privacy risks.

This study examines the claim that membership inference attacks can be used to quantify privacy gains from using various privacy-preserving techniques. The scenario modeled in these experiments simulates the situation where the privacy-preserving technique is perfect pseudonymization. Every datapoint with a named patient in the training data for ClinicalBERT has a corresponding datapoint in the *Real* dataset where the name is different. In such a scenario, no real names are left in the training data to memorize. Thus, the risk of leaking any name of a patient is zero, represent-

ing a substantial increase in privacy. If the attack accurately quantifies these privacy gains, then we would expect it to perform worse when the data has been pseudonymized.

## 3 Results

Three different attacks are performed using three different datasets as the in-data. The accuracy, precision, and recall values of each attack are listed in Table 1. Experiment (A) mirrors the setup used by Mireshghallah et al. (2022). Experiments (B) and (C) use the subsets of MIMIC-III that only contain names. There are only very small differences in the correctness of the classifications, regardless of the configuration used.

Table 1 also lists the AUC, which represents a threshold-independent evaluation of the MIAs. The AUC varies more than the other three metrics. However, the difference between experiments (A) and (B) is larger than that between experiments (B) and (C). This is despite the fact that the *In-data* for experiments (A) and (B) come from the same population. The difference in AUC between experiments (B) and (C) is 0.017.

Experiments (A) and (B) represent cases where we have not performed any pseudonymization of the training data. That is, the *In-data* are used to train the BERT model without employing any privacy-preserving techniques. Experiment (C) is the result of the simulated scenario where perfect pseudonymization is employed to preserve the privacy of the data. In other words, the model is not exposed to any real names during training. The privacy gains from using this technique are not reflected by the metrics in Table 1.

| | In-data | Out-data | Threshold | Accuracy | Precision | Recall | AUC |
|---|---|---|---|---|---|---|---|
| (A) | *Pseudo, random sample* | *Held-out* | *i2b2* | *0.771* | *0.990* | *0.548* | *0.916* |
| (B) | Pseudo, names only | Held-out | i2b2 | 0.780 | 0.990 | 0.566 | 0.882 |
| (C) | Real, names only | Held-out | i2b2 | 0.770 | 0.990 | 0.548 | 0.865 |

Table 1: The membership inference attack is run with three different configurations. Experiment (A) uses a random sample of MIMIC-III used in Mireshghallah et al. (2022) as in-data, and all experiments use the same out-data as they do. Experiments (B) and (C) use the datasets described in Section 2.1 for the in-data. The accuracy of each attack is displayed alongside the recall and the precision values. The threshold-independent AUC value is also listed.

## 4 Discussion and Conclusions

This study focuses specifically on protecting names. Future research would benefit from analyzing additional categories of PII. However, the data and models created by Lehman et al. (2021) focus specifically on names. This class of PII is used in this study to facilitate comparisons with earlier studies.

The results from the three experiments in Table 1 are very similar to each other. At the same time, experiment (C) represents a scenario in which a very strong privacy-preserving measure has been employed to increase the privacy of the target model. If the studied MIA is an accurate way of quantifying the privacy benefits of using pseudonymization, then we would expect the MIA to be much less accurate in experiment (C). The fact that the MIA works nearly as well for experiments (A) and (B) as for (C) indicates that using this attack to quantify memorization does so on a datapoint level. This may be useful for evaluating techniques such as differentially private pretraining (Li et al., 2022), which operate on entire datapoints.

It remains to be shown which of the datapoint's characteristics are used to separate members from non-members. The results of our experiments suggest that using this MIA does not accurately quantify the privacy gains from using pseudonymization, which instead operates on the token level. While the scope of this short paper was limited to evaluating a state-of-the-art MIA for BERT models, future research should also evaluate other MIAs and a wider range of privacy-preserving techniques.

## Acknowledgements

We want to thank Fatemehsadat Mireshghallah for sharing the data and code used in Mireshghallah et al. (2022). We are also grateful to the DataLEASH project for funding the research presented in this paper.

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
