# OpenReview forum: "Using Membership Inference Attacks to Evaluate Privacy-Preserving Language Modeling Fails for Pseudonymizing Data"
_NoDaLiDa/2023/Conference — NoDaLiDa 2023_

### Official Review · Reviewer_RZgo · 2023-03-11
**focused contribution on privacy-enhancing NLP**

**Rating:** 7
**Confidence:** 4

**Review:**

The paper presents experiments showing that membership inference attacks fail to detect any privacy benefit from pseudonymizing text data, although we would intuitively assume that such pseudonymizing efforts would at least provide some privacy protection compared to the raw texts.

The paper is clear and well-written, although the results are perhaps not too surprising. After all, membership inference attacks seek to detect whether a particular data point (in this case, a text) has been used to train or fine-tune a given model. Although the replacement of names with their pseudonyms might remove a few cues, the text is still likely to contain many phrases/patterns (or combinations of those) that are unique to that particular document.

One shortcoming of the approach is that the pseudonymisation is only done on names (I assume that "names" are here used as synonym for "person names"). It would be useful to see if the results change if one were to conduct a full de-identification (including e.g. all named entities, dates, identification numbers, etc.).

On a more general note, it would be worth mentioning that membership inference attacks and pseudonymization (I personally prefer the terms text de-identification or sanitization) seek to address slightly different problems. Pseudonymization seeks to make it harder to single out an individual from the data, notably based on assumptions about the background knowledge that may be available to an adversary. This may also provide at least some protection about inferring whether a given person is mentioned / referred to in the text collection. Membership inference attacks such as the ones presented by Mireshghallah (2022), on the other hand, seek to assess whether one can detect the use of a *datapoint* (in this case, a text) in a ML model.  These are two relatively distinct membership inference problems, as the experimental results suggest.

**Paper Type:**

Short paper

---

### Official Review · Reviewer_cAN4 · 2023-03-13
**Focused short paper with a solid but limited experiment**

**Rating:** 7
**Confidence:** 4

**Review:**

The paper presents a short focused study in which a model passes a membership inference attack, a test whether a model reveals whether a certain data-point has been used to train it. This is accomplished by training a target model on the target data and a reference model on similar data and then the task of the attack is to compare loss of the models on the examples to decide whether they have been seen by the target model - its loss in such case should be lower than the loss of the reference model. Precision and recall are measured on classifications of instances to fit the target model. The models and the datasets are taken from an existing study which serves as a baseline. In addition two datasets are artificially created that only contain sentences with names which are swapped in pseudonymisation steps. In the last experiment the model is also trained on the pseudonymised data. The results indicate no difference between all configurations and the authors conclude that MIA does not detect pseudonymisation and conclude that memorisation must operate on an instance/token level.

This is expected since the embeddings are contextualised embeddings and BERT learns contexts and not individual words. Work has been done on probing language models and in particular evaluation evaluation of the NLI (Natural Language Inference) task which has shown that models are not using lexical information in order to make inference which is also reflected in the findings here. For example,

A. Talman, M. Apidianaki, S. Chatzikyriakidis, and J. Tiedemann. How does data corruption affect natural language understanding models? a study on GLUE datasets. In Proceedings of the 11th Joint Conference on Lexical and Computational Semantics, pages 226–233, Seattle, Washington, July 2022. Association for Computational Linguistics.

The authors do not make a stronger case but pseudonymisation based only on swapping names is clearly on over-simplification of the task since humans too are likely to be able to make a good guess using their knowledge of the context of words in a target sentence and their memories where they have heard sentences to connect to connect them to particular situations even if the name is removed or swapped. They may be able to connect a particular statement to a person even if a name in a statement is swapped (experiment C). This points out that the task described here is too simple.

**Paper Type:**

Short paper

---

### Decision · Program_Chairs · 2023-03-17

Accept